# Landscape Efficiency Assessment of Urban Subway Station Entrance Based on Structural Equation Model: Case Study of Main Urban Area of Nanjing

**Zhe Li *** , **Xiaoshan Lin, Xiao Han, Xinyi Lu and Hengyi Zhao**

School of Architecture, Southeast University, Nanjing 210096, China; 220190210@seu.edu.cn (X.L.);
hanxiaoseu@seu.edu.cn (X.H.); luxinyi@seu.edu.cn (X.L.); zhaohengyi@seu.edu.cn (H.Z.)
* Correspondence: lizheseu@seu.edu.cn

**Abstract:** Public landscape efficiency is one of the research hotspots in contemporary landscape performance. The renewal of micro landscape space has positive effects on community vitality and the sustainable development of landscape resources. The subway station entrance is a typical representative of the miniature landscape environment. To improve the construction of subway station entrances, clear cognition on the landscape efficiency of subway station entrances and their impact indicators is necessary. For this purpose, a structural equation model with a parameter system was established to measure landscape efficiency. There are four latent variables (e.g., traffic capacity) and 10 observed variables (e.g., visual level) composed of an estimation model. Researchers selected 131 subway station entrances in the main urban area of Nanjing as survey samples. Various methods, including investigation, image recognition, and modelling analysis, were comprehensively used to analyse the landscape characteristics of the subway station entrances quantitatively. A calculation was conducted to obtain the correlation coefficient of latent variables and the explanatory degree of observed variables. The result shows that green space landscapes and traffic capacity impresses the landscape efficiency of subway station entrances. Furthermore, all these variables have complex correlations. The fluctuation of any latent variable may cause the decay or enhancement of related variables. Therefore, designers should have a comprehensive cognition of the landscape environment of subway station entrances to enable them to propose balanced design strategies under traffic, visibility, plants and facilities. This paper aims to help designers gain an in-depth understanding of the ideal landscape construction forms at subway station entrances and facilitate the high-quality development of the urban landscape environment.

**Keywords:** structural equation model (SEM); latent variable; path analysis; landscape efficiency; subway station entrance





## 1. Introduction

Contemporary landscape research has gradually shifted from the intuitive perception and empiricism of qualitative research to scientific research and quantitative analysis, and research actively carried out on the sustainable utilization and renewal of urban public landscape resources [1].

Landscape renovation of miniature public spaces has become the focus of urban high-quality development research and practice. Researchers have conducted continuous investigations and explorations of urban micro-regeneration based on scientific cognition and quantitative evaluation [2]. Behind it is the urgent need for landscape architecture in response to the challenges of urbanization and to improve the technology of design decision-making [3].

The urban subway station entrance is an important branch of small-scale public space. As a landscape hub of rail transportation, it can maintain coordination with urban space nearby and bring generous landscape experience for residents [4]. However, the landscape

potential of subway station entrances has been critically ignored in previous constructions. Many subway station entrances are isolated, restrained and pedestrian-unfriendly due to a lack of overall design and planning [5]. We were therefore urged to scientifically analyse the landscape composition forms and existing deficiencies of subway station entrances conducting a large sample survey to propose universal design principles and strategies for landscape renewal.

Landscape efficiency is a measure of the effectiveness of landscape design practice to achieve the expected objectives under the premise of sustainable development [6]. Its primary principle is to maximize the comprehensive benefits comprising the environment, economy and society [7]. Previous studies frequently concentrated on making longitudinal performance comparisons in small-scale public projects and have used various algorithms [8] to analyse the connection between individual elements and overall landscape experience. While a lack of quantitative data collection is the major deficiency of landscape efficiency research, with the development of big data technology, the acquisition of urban environmental datasets has been conducive to realize the mathematical description of landscape efficiency [9].

Regarding the landscape efficiency of subway station entrances, its dynamic fluctuation is related to periodic pedestrian volume. There are several landscape elements that directly affect the subway station entrance environment, some of which are abstract and complex in theory. In general, these elements are composed of a set of similar independent factors, which are non-observed factors or latent variables of landscape efficiency. In factor analysis, structure simplification of latent variables is an indispensable procedure to solve estimation problems [10]. As an effective multivariate statistical method, a structural equation model can verify the theoretical hypothesis relationship and the path loading between latent variables by setting a covariance matrix [11]. Compared with the SEM method, qualitative analysis such as questionnaires and case studies are still insufficient in the selection of evaluation indicators and verification of proposed evaluation system [12]. The SEM method can help researchers to improve model production efficiency and enhance the ability in response to academic controversies by providing a statistically verifiable framework and a graphical model [13]. At present, the SEM methods are widely used in performance investigation, such as public space governance and visual quality evaluation [14,15].

This paper aims to extend the technical application of a structural equation model in the landscape field and propose a method for realizing the quantitative evaluation of the landscape efficiency at subway station entrances. Researchers preliminarily focus on the overall landscape characteristics of subway station entrances, select related impact factors into categories based on the literature review, and construct an evaluation system for landscape efficiency. The main urban area of Nanjing was selected as the research area. The experiments were conducted by inputting the measured data and observing the model fitting situation. The results, including a quantitative description and graphic expression, analyse the ideal landscape morphology, and propose the corresponding design strategies of the subway station entrances, which have reference significance for the renovation of similar miniature public spaces.

## 2. Relevant Research Progress

### 2.1. Study on Subway Station Entrance Landscape

The subway station entrance landscape is a distinctive urban open space [16], which has the characteristics of traffic-orientation location, comfortable accessibility and diverse landscape elements. In this field, investigations are concentrated on discussing the influences of built environments around subway station entrances [17]. Brovarone (2020) discussed how public transport stations play their role as potential landscape elements, shaping urban environmental characteristics [18]. Li et al. (2018) pointed out that the urban rail transit landscape corridor composed of subway stations has a regional integration effect on the city [19]. Some studies have considered the construction of a landscape evaluation system for subway station entrances and have gained various datasets of environment

characteristics. Sun et al. (2017) collected 756 micro scale environmental data and proposed a tool to measure the walking environment around rail transit [20]. In addition, pedestrian traffic patterns become another research highlight [21]. Shirane et al. (2016) studied a method to develop the urban environment around the subway station entrance, which helps to ensure the smooth flow of pedestrians and enhances the vitality of neighbourhood communities [22].

However, previous research has underestimated the application of digital technology, resulting in difficult and time-consuming data collection. There are still a dearth of effective verification methods for an evaluation system of the subway station entrance landscape environment. This paper attempts to optimize the research approach by introducing structural equation models to strengthen theoretical testing validity. Meanwhile, digital measurement, such as image recognition and statistical modelling, can enhance quantitative evaluation and improve data acquisition efficiency.

### 2.2. Research on Structural Equation Model

A structural equation model is a confirmatory statistical analysis method to simulate the multiple causality of potential factors [23], which is usually composed of observed variables and latent variables [24]. Its theoretical core is to transform the causality relationship between variables into an objective mathematical matrix by observing the matching degree between the theoretical model and empirical data.

For the current study, the application of SEM mainly focuses on the research fields of correlation analysis, evaluation system and performance research. In recent advances, Curl and Mason (2019) demonstrated a correlation between urban landscape, pedestrian space and mental health by constructing a latent causal path modelling [25]. Typical public landscapes such as scenic spots and urban parks have become hotspots in the field of SEM confirmatory research [26]. It is the focus of previous research to deduce the user's subjective preference for the landscape environment, verifying the questionnaire results, such as users' travel motivation, behaviour intention of recreational activities and public satisfaction [27]. Meanwhile, multi-index evaluation models of miniature-built environments have advanced [28]. A landscape efficiency evaluation model based on parametric spatial element extraction was proposed to explore the potential relationship among the observable landscape elements [29].

With the technical innovation of the structural equation model, Amini and Alimo-hammadlou (2021) developed the equation structural model (ESM) based on interpretive structural modelling (ISM), which eliminates the reliance on participants' intuition [30]. The model generating strategy improved the analysis efficiency of the evaluation model [31]. In addition, the integrated application of multiply digital technology improves the objectivity and measurability of sample data [32,33]. Said et al. (2017) proposed two models for landscape satisfaction evaluation and indicated particular neighbourhood attributes which have great impacts on the pedestrian environment [34]. Wang et al. (2019) investigated Helsinki with a Public Participatory Geographic Information System (PPGIS) and estimated the correlations between seven environmental factors and four biodiversity indicators [35]. Nicolas et al. (2021) quantified the management dynamics and the performance objectives completion of the smart city with the help of the structural equation model (SEM) [36].

With the big data era, there has been a technical revolution that improves data integration, processing, and interaction. However, the application of the structural equation model in landscape efficiency is still scarce and unsatisfactory. The major problem is to overwhelmingly emphasize the participants' perception instead of objective landscape characteristics in quantitative evaluation. Previous studies have neglected the integrity and complexity of landscape environment, despite some which have discovered the efficiency contribution from particular landscape elements. To enhance the SEM method, this paper adopts a more complex multivariate model structure in order to realize a hierarchical and intuitive path coefficient description among latent and observable factors. Further studies

are still essential to obtain a comprehensive measurement of landscape efficiency in other small-scale built environments under large-sample analysis.

## 3. Theoretical Model of Urban Subway Station Entrance Landscape Efficiency

### 3.1. Logical Framework

Subway station entrances emphasize pedestrian functions and take traffic efficiency as the primary principle in path organization. The facilities need to meet safety and humanization demands while harmonizing with the dynamic traffic conditions [37]. It is appropriate to build a green space with rich landscape layers and high visibility around subway station entrances. In order to measure the landscape efficiency of subway station entrances, the observable elements with direct influence are preliminarily analysed and classified.

From the literature review it can be seen that four categories: traffic function, visual perception, landscape facilities and green space, mainly affected the landscape efficiency of a subway station entrance. The walkability of subway station entrances is significantly related to the road network pattern, spatial boundary, and enclosure structure [38,39]. The visual openness of landscape space dramatically influences the visual perception of users and indirectly affects users' satisfaction and safety assessment in the subway station entrance environment [40]. Landscape facilities in the subway entrances include lighting, seats, pavement, barrier free facilities, shared bicycle parking spaces, etc. Quality, quantity and distribution characteristics are used as the evaluation principles to measure their landscape benefits. Green space can enrich the aesthetic experience of pedestrians and improve the outdoor microclimate [41]. The common evaluation factors of public green space are mainly related to planting pattern, canopy shade and vegetation structure [42].

This study excludes the weakly related factors of the subway station entrance landscape environment and contemplates the need for data clustering and quantitative measurement difficulties. As illustrated in Table 1, four indicators (traffic capacity, visual openness, green space landscape and service capacity of landscape facilities) and ten factors (e.g., connectivity, visual level) were selected to characterize the landscape efficiency of subway station entrances. According to the field research and literature review, it is speculated that all indicators have positive effects on landscape efficiency to a varying degree, and the selected factors can feature corresponding indicators.

Specifically, as the structural characteristics of subway station entrances, traffic capacity and visual openness can reflect the macroscopic landscape morphology and directly affect landscape efficiency. The green space landscape and the service capacity of landscape facilities are made up of a collection of relevant factors. These two indicators exert specific service functions and have multiple effects on landscape efficiency. Besides the factor set of landscape efficiency, street interface permeability is introduced into the evaluation system as a restrictive variable to limit the enhancement of indicators.

### 3.2. Technical Route

As shown in Figure 1, there are five steps to build a landscape efficiency evaluation model of subway station entrances. This paper is of significant importance to landscape efficiency improvement and the landscape transformation of subway station entrances. The research method can be applied to similar small-scale landscape projects, such as the green space of streets, pocket gardens, and public squares.

**Table 1.** Factor set of subway station entrance landscape efficiency evaluation system.

| Indicator | Factor | Explanation | Method | Unit |
|---|---|---|---|---|
| Traffic capacity | Connectivity | It reflects the connection degree among different paths. Pedestrian usually enter a space with higher connectivity. | CAD; Depthmap | / |
| Visual openness | Visual level | It reflects the visual conditions of any target point in the landscape space. The higher the visual level, the easier the space is to be observed and perceived. | CAD; Depthmap; field survey | / |
| Evaluate Factors Green space landscape | Average visual green quantity | It reflects the average level of green quantity perceived by visitors when entering the subway station entrances. | Photoshop; semantic segmentation | / |
| | Vegetation canopy green quantity | It reflects the average level of vegetation canopy structure in the walking environment of subway station entrances. | Photoshop; semantic segmentation | / |
| | Plant community species density | It reflects the richness of plant species on the site. (i.e., the ratio of plant community species number to the site area.) | / | Species/m² |
| | Shading rate of walking space | It reflects the shading capacity of plants, indicating the ability of trees to provide a comfortable thermal environment. | Field survey; CAD | / |
| Service capacity of landscape facilities | Density of recreational facilities | It reflects the capacity of recreational activities. (i.e., the ratio of the number of seats to the site area.) | Field survey; area measurement | pieces/m² |
| | Accessibility coverage | It reflects matching degree between barrier free facilities available and barrier free facilities required. | Area measurement | / |
| | Pavement integrity of pedestrian space | It reflects the construction quality of walking space. The higher pavement integrity is, the higher the pedestrian satisfactory might be. | Field survey | / |
| | Bearing capacity of parking facilities | It reflects the visiting and usage demand of residents. (i.e., ratio of the number of non-motor vehicle parking spaces to the site area.) | Field survey; area measuremen; | pcs/m² |
| Limiting factor | External environment Permeability of street interface | It reflects the spatial recognition of the entrance environments from the street interface. | CAD | / |

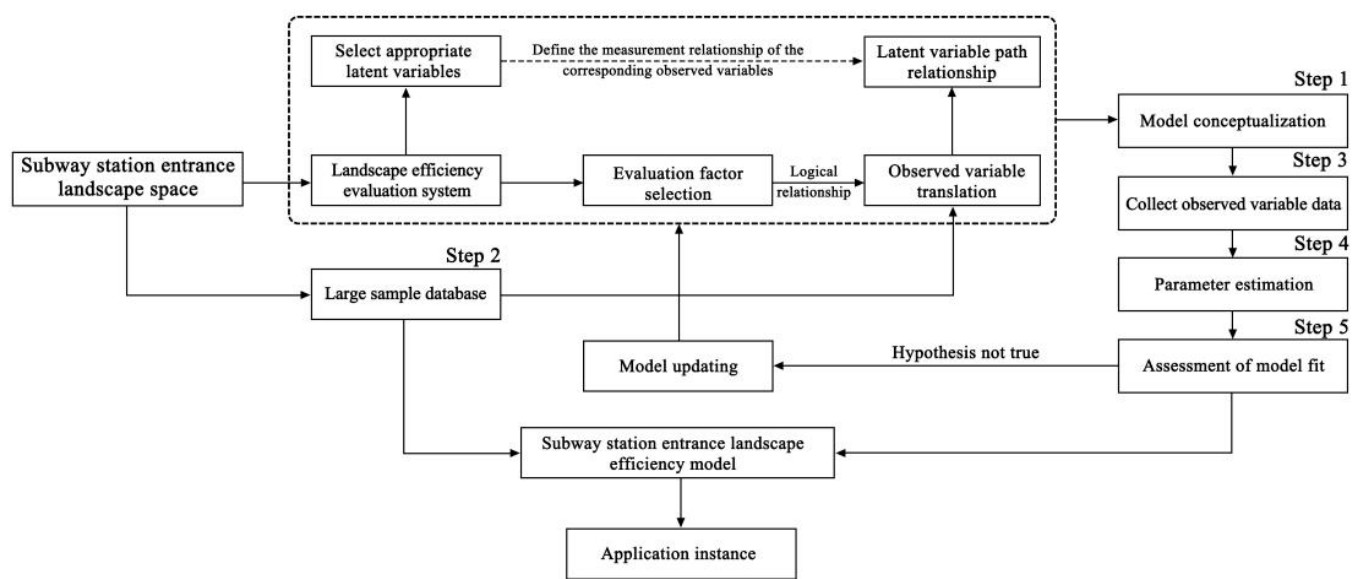

**Figure 1.** Steps of constructing landscape efficiency evaluation model.

(1) Establish a theoretical model: Transform the proposed evaluation system into a structural equation model. The indicator layer corresponds to the latent variables which cannot be extracted directly from the landscape environment. The factor layer is equivalent to the observed variables. Then, researchers set the causal relationship

among latent variables and measure the relationship between latent variables and observed variables individually.

(2) Construction of a sample database: Select suitable samples according to site size and space boundary. The property line of the subway station entrance restricts the sample area, excluding the adjacent sidewalks and the municipal public green space.

(3) Data collection and processing: Collect observed data of each sample, then batch process the original data to eliminate the effects of different dimensions. Since the maximum and minimum values of the data are known and there is no obvious outlier data, the "min-max standardization" method is suitable to be used in this study.

(4) Parameter estimation: Import the processed data into the model and build the mathematical matrix for the regression model. After the parameters reach a single value, it shows that the evaluation system and theoretical hypotheses exist.

(5) Fitness evaluation and model modification: Comprehensively evaluate the model validity regarding different fitness indicators. When the theoretical model does not match well, the initial model must be locally adjusted and optimized. The evaluation model is successfully established after completing the overall verification.

### 3.3. Initial Model and Theoretical Hypotheses

According to the evaluation system of subway station entrance landscape efficiency, an initial model was constructed with logical paths. As shown in Figure 2, the initial theoretical model consists of five latent variables and 11 observed variables, which are divided into the structural model and the measured model. The part used for the description of relationship between latent variables is regarded as the structural model. In contrast, the measured model is used to measure the latent variables by setting corresponding observed variables.

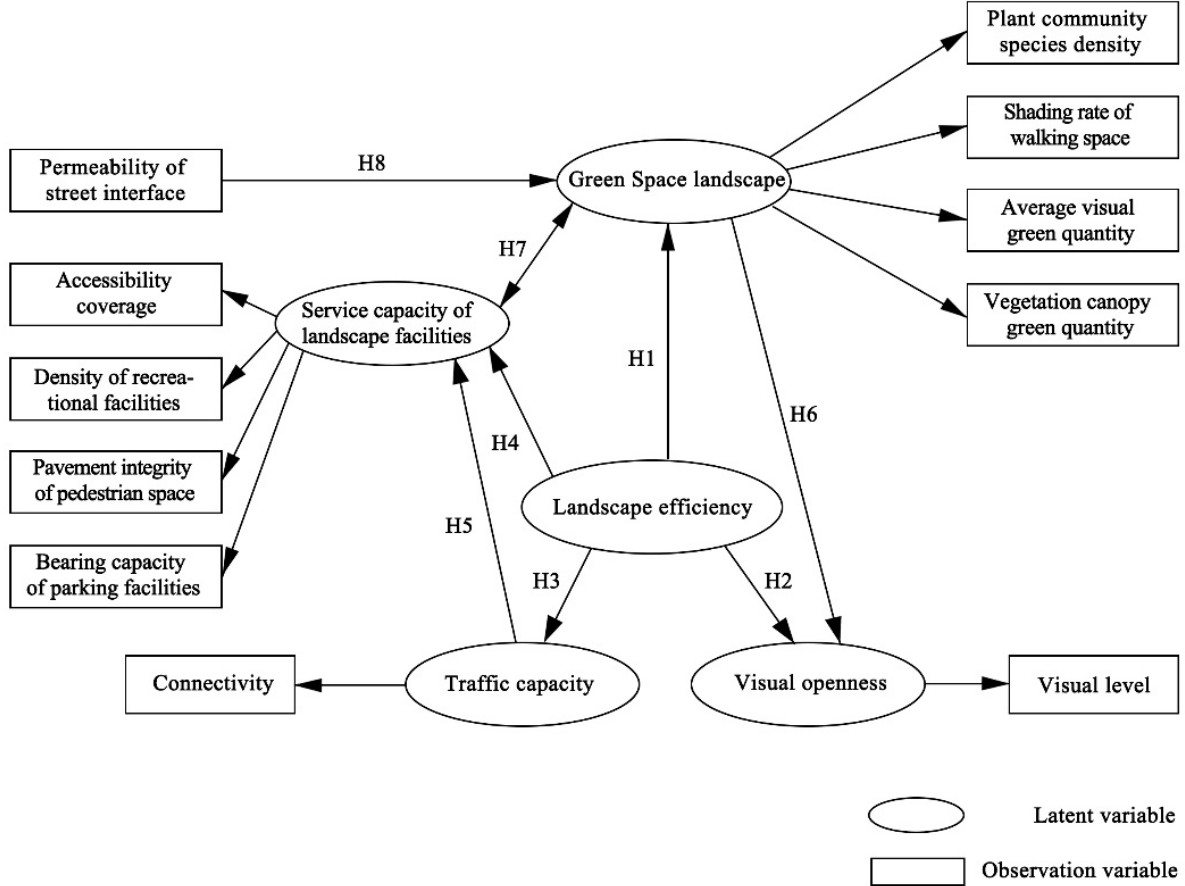

**Figure 2.** Initial model and theoretical hypotheses of the landscape efficiency of subway station entrances.

The composition relationship of the measurement model is relatively clear. Green space landscape can be interpreted by the four observed indicators (plant community species density, shading rate of walking space, average visual green quantity and vegetation canopy green quantity). The measurement of the service capacity of landscape facilities is derived from four variables (density of recreational facilities, accessibility coverage, pavement integrity of pedestrian space and bearing capacity of parking facilities). However, traffic capacity and visual openness contain only one observed variable: connectivity and visual level, respectively.

As the following eight research hypotheses illustrate, the structural model is composed of five latent variables with a relatively complex path effect.

**Hypotheses 1–4 (H1–H4):** *The latent variable "landscape efficiency" is directly influenced by four latent variables: green space landscape, landscape facility service capacity, visual openness and traffic capacity.*

**Hypotheses 5 and 6 (H5 and H6):** *Path connectivity directly affects landscape facility service capacity, while green space landscape may have a moderating effect on visual openness.*

**Hypotheses 7 and 8 (H7 and H8):** *There is a mutual relationship between green space landscape and landscape facility service capacity, including unidirectional promotion and inherently restriction. Meanwhile, the permeability of the street interface has a negative effect on green space landscape.*

## 4. Case Study

### 4.1. Overview of Studied Area and Sample Library

At the end of 2020, Nanjing completed the construction of 10 subway lines with 174 stations, making it the first city in China to open subways in all districts and counties. Subway construction there has promoted the optimization and development of the urban spatial pattern, driven the intensive development of new urban areas, improved traffic congestion in the downtown area, and provided convenient travel conditions for inter-district traffic.

The main urban area is the core functional zone of Nanjing and it is also the area with the highest intensity of subway construction. At present, the railway lines distributed in the main urban area of Nanjing include Line 1, 2, 3, 4, 10 and S3, comprising 95 subway stations (including repeated transfer stations), of which 78 are underground stations, accounting for 82.1%. There are 320 subway station entrances in the main urban area, of which 230 are independent. It was investigated that the independent subway station entrances are located around main roads, close to residential areas, commercial plazas, city parks and core scenic spots. Their scales are concentrated in 400–1000 m², which can basically meet the traffic needs of residents. However, there are some landscape problems at these subway station entrances, such as monotonous landscape form, right-of-way conflicts, high parking density, inadequate landscape facilities, and so on. Owing to the large quantities of subway station entrances in the downtown area, Nanjing has full potential to be regarded as the survey area.

For the construction of the database, it was suggested to select over 100 samples for convergence degree and parameter stability [43]. Considering the construction situation of the existing independent subway station entrances in Nanjing, we finally selected 131 entrances as the survey samples, including Gulou subway station and entrance 6 of Jimingsi subway station (Figure 3).

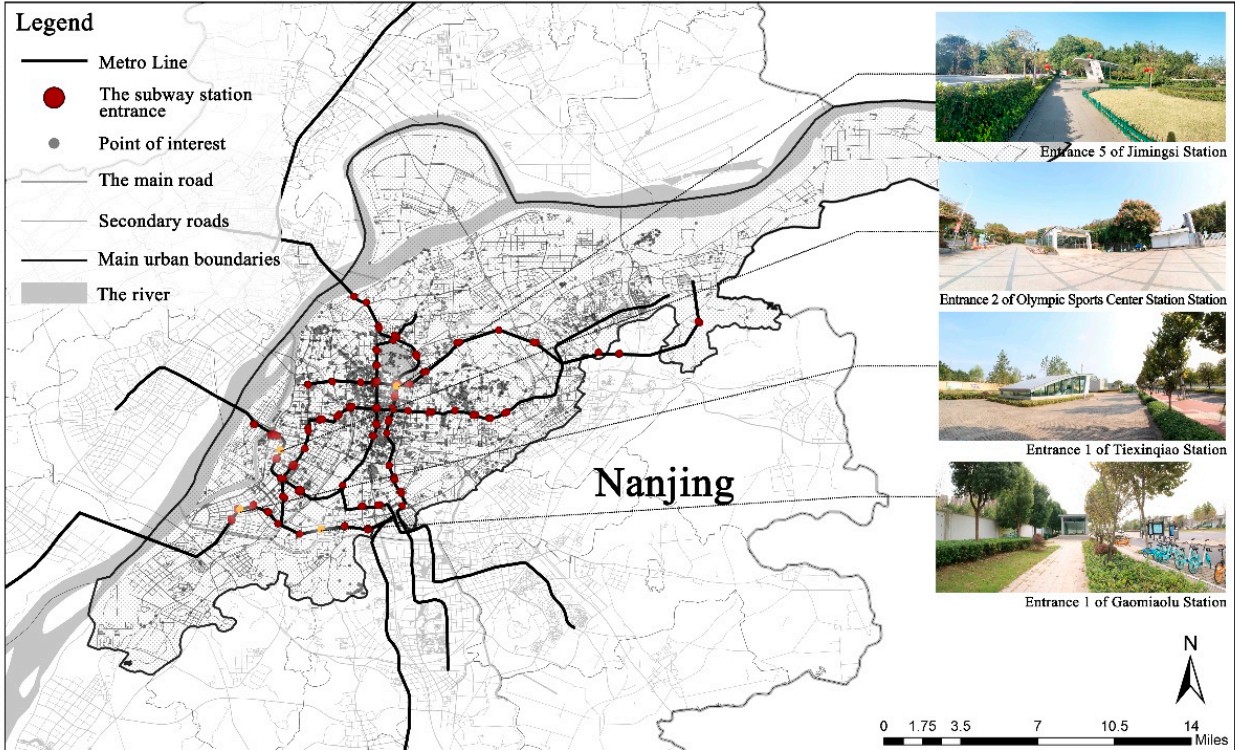

**Figure 3.** Sample case distribution map of subway station entrances in the main urban area of Nanjing.

### 4.2. Data Collection and Pre-Processing

As shown in Table 2, this paper conducted a combination of field survey, image recognition and modelling to enhance efficiency and accuracy in the process of data collection.

The latent variable "landscape facility service capability" reflects the density, distribution of facilities in the landscape environment. The subordinate observed variables were obtained by measuring spatial elements through field surveys, photography and some other methods.

The observed variables of latent variables, "traffic capacity" and "visual openness", are connectivity and visibility level, respectively, both of which reflect the spatial organization efficiency of subway station entrances. In data collection of connectivity, it is the step that draws the accessible space of the site in the dxf format and establishes an axis model to calculate the connectivity score (Figure 4). The acquisition of visibility level is similar to the previous variable. The difference is that the invisible occlusion areas are drawn before the construction of the VGA model (Figure 5).

The latent variable "green space landscape" is composed of four observed variables. Basically, the data collection for these variables relies on photography and measurement. After taking photos on site, both the average visual green quantity and vegetation canopy green quantity apply the semantic segmentation technology to recognize the proportion of green space pixel values in each photo (Figure 6). The photographic equipment includes SLR camera, wide-angle lens and fisheye lens [44]. It is clear that the performance of these two factors has great differences resulting from distinct greening density of the subway station entrances (Figures 7 and 8). After data collection, it is necessary to import original data into SPSS software for normalization, so as to eliminate the influence of different dimensions.

**Table 2.** Collection methods of observed variable data.

| Observation Variable | Calculation Formula | Explanation | Data Collection Method | Data Source/Instrument |
|---|---|---|---|---|
| Connectivity | $X_1 = \frac{1}{RAA}$ | $RAA$ is the global depth of the axis model. | (1) Import the site plan in dwg format into CAD, retain accessible spaces such as roads and squares, and convert it to dxf format. (2) Import the file into Depthmap, build the axis model, and calculate the integration coefficient. | Nanjing Municipal Bureau of Land and Resources |
| Visual level | $X_2 = \text{card}(M)$ | $G$ represents the total number of grids. For any target point S, all observation points that can be seen are "$M$". | (1) Measure the position and canopy width of trees below 1.5 m in height. (2) Draw a site plan in CAD, including the visible elements, such as entrance pavilions and plants. (3) Import the file into Depthmap, establish a visibility model with the grid size in units of 0.6 m, and calculate the visual integration. | Nanjing Municipal Bureau of Land and Resources Tape measure |
| Average visual green quantity | $X_3 = \frac{\sum\limits_{n}^{i=1} \frac{P_g}{P_t}}{n}$ | $P_g$ represents the amount of green pixels in the "ith" panorama, and $P_t$ represents the total number of pixels. | (1) Simulate the process of entering the stations from outside, and determine at least 3 observation points within 250 m. (2) Set the camera in road centreline at the height of 1.5 m and take photos. (3) Calibrate of the image distortion and recognize the proportion of green space pixel values of each photo. | Wide-angle lens; SLR camera |
| Vegetation canopy green quantity | $X_4 = \frac{\sum\limits_{n}^{i=1} \frac{S_g}{S_t}}{n}$ | $S_g$ represents the amount of green space pixels in the "ith" photo, and $S_t$ represents the total number of pixels. | (1) Select 3–5 observation points with differences (2) Fix the camera at a height of 1.5 m and take photos in sky photography mode. (3) Recognize the proportion of green space pixel values. Note: We conducted the survey from June to September 2021 on cloudy days with weak wind speed. | Fisheye lens; SLR camera |
| Plant community species density | $X_5 = \frac{S}{\ln A}$ | $S$ represents plant species and $A$ is the sample area. | Calculate the varieties of plants and introduce the number into the formula. | / |
| Shading rate of walking space | $X_6 = \frac{S_{ts}}{S_w}$ | $S_{ts}$ is the projection area of trees in the pedestrian space, and $S_w$ is the total area of the pedestrian space. | (1) Measure the location and canopy width of trees and draw in the CAD. (2) Measure the projection area of trees and the total area of the pedestrian space. | / |
| Density of recreational facilities | $X_7 = \frac{N_s}{S}$ | $N_s$ is the number of seats, and $S$ is the space unit area. | Investigate the number of seats, and introduce it into the formula for calculation | / |
| Accessibility coverage | $X_8 = \sum \frac{AA_{na}}{SA_{na}}$ | $AA_{na}$ represents the quantity of facilities available, and $SA_{na}$ is the quantity required. | Classify and count the number of barrier-free facilities such as ramps, escalators, straight ladders, blind lanes and signs through field research. | / |
| Pavement integrity of pedestrian space | $X_9 = \frac{O_{nl} - I_{nl}}{O_{nl}}$ | $O_{nl}$ is the total area of paving space, and $I_{nl}$ is the total area of paving space in flaws. | Estimate or measure the complete paving area of the pedestrian space (excluding damage, cracks, warping, breakage, etc.) | Tape measure |
| Bearing capacity of parking facilities | $X_{10} = \frac{\frac{S_{nmv}}{1.2}}{S}$ | $S_{nmv}$ is the parking area of non-motor vehicles, and $S$ represents the total area of the site. | Measure area of parking facilities. The number of parking space is estimated according to 1.2 m² /a single parking space. | Tape measure |
| Permeability of street interface | $X_{11} = \frac{\sum\limits_{t=1}^{n} a_i * t}{S}$ | $t$ is the penetration of interface i, a is the area of interface i, and $S$ is the total vertical interface area. | Parallel to the road, take the street view of subway station entrances. Measure the areas of different vertical interfaces. | SLR camera |

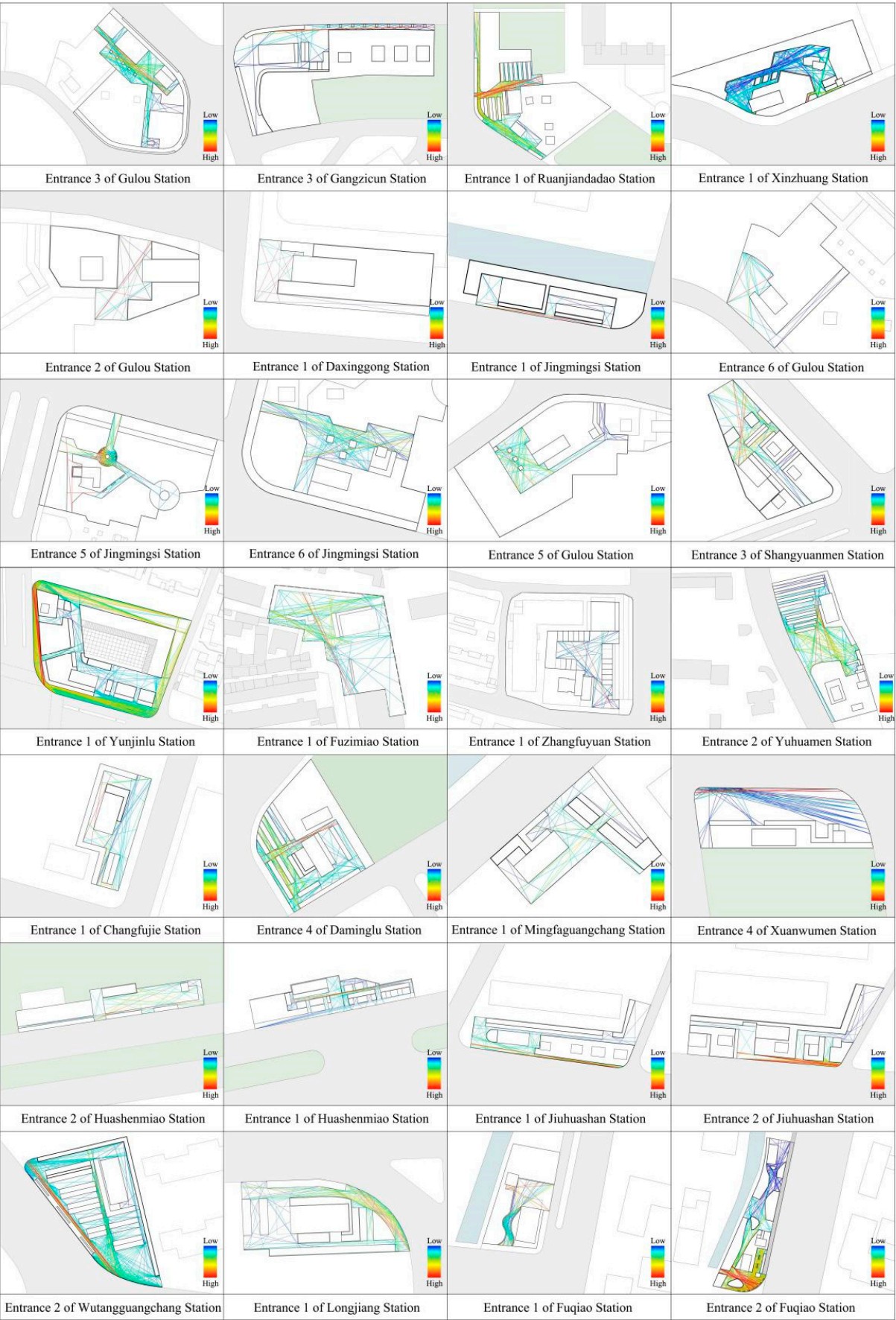

**Figure 4.** Examples of acquisition results of traffic capacity.

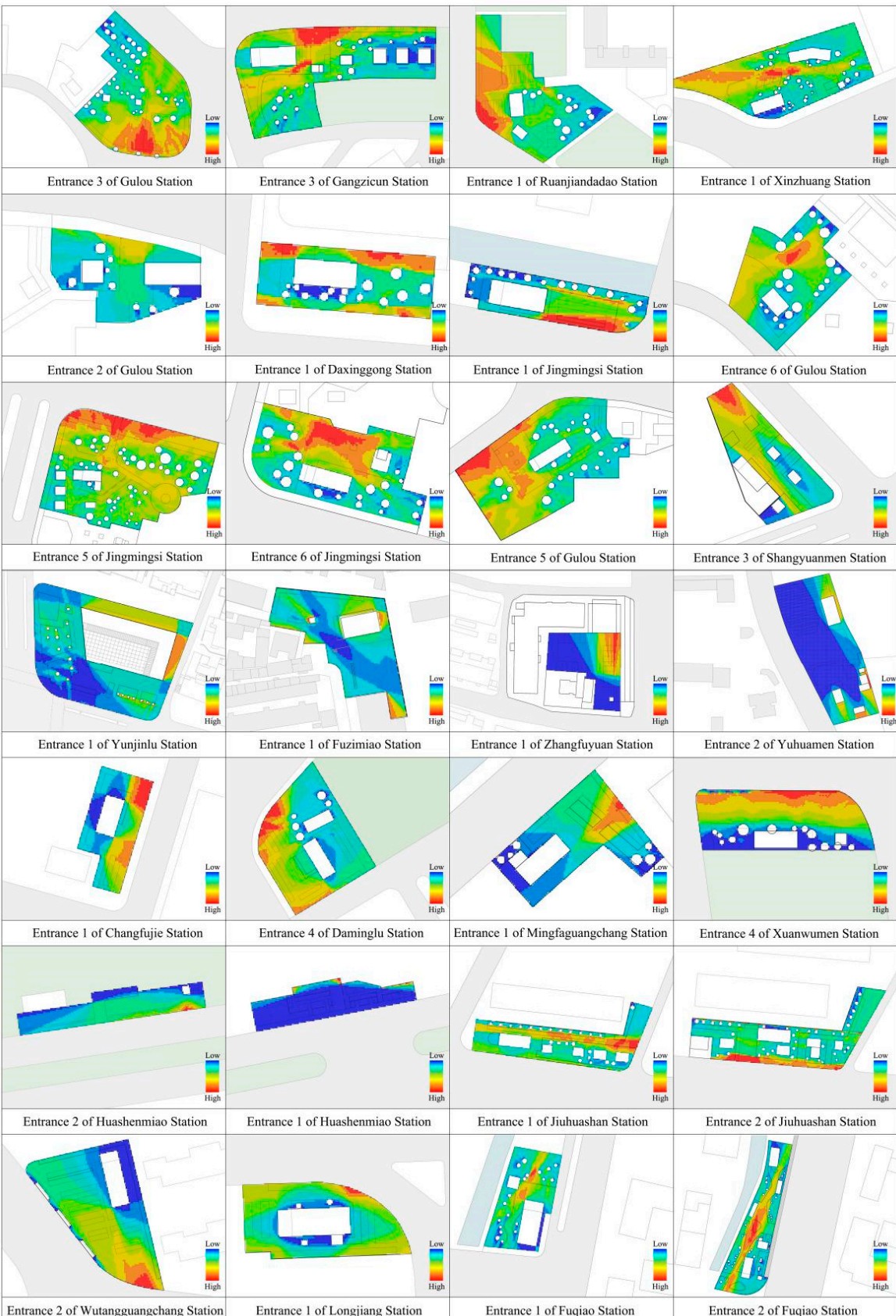

**Figure 5.** Examples of acquisition results of visual openness.

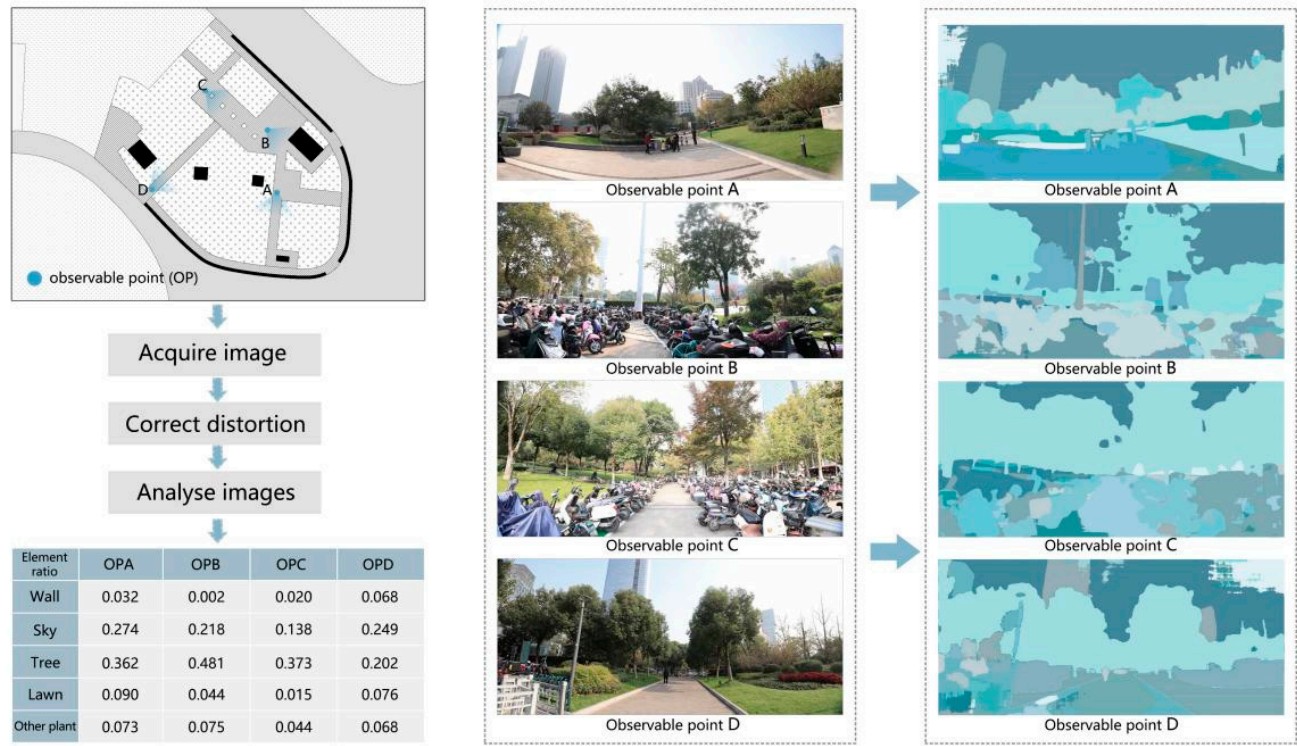

The table shown in the figure:

| Element ratio | OPA | OPB | OPC | OPD |
|---|---|---|---|---|
| Wall | 0.032 | 0.002 | 0.020 | 0.068 |
| Sky | 0.274 | 0.218 | 0.138 | 0.249 |
| Tree | 0.362 | 0.481 | 0.373 | 0.202 |
| Lawn | 0.090 | 0.044 | 0.015 | 0.076 |
| Other plant | 0.073 | 0.075 | 0.044 | 0.068 |

**Figure 6.** Schematic illustration of the collection method of the average visual green quantity.

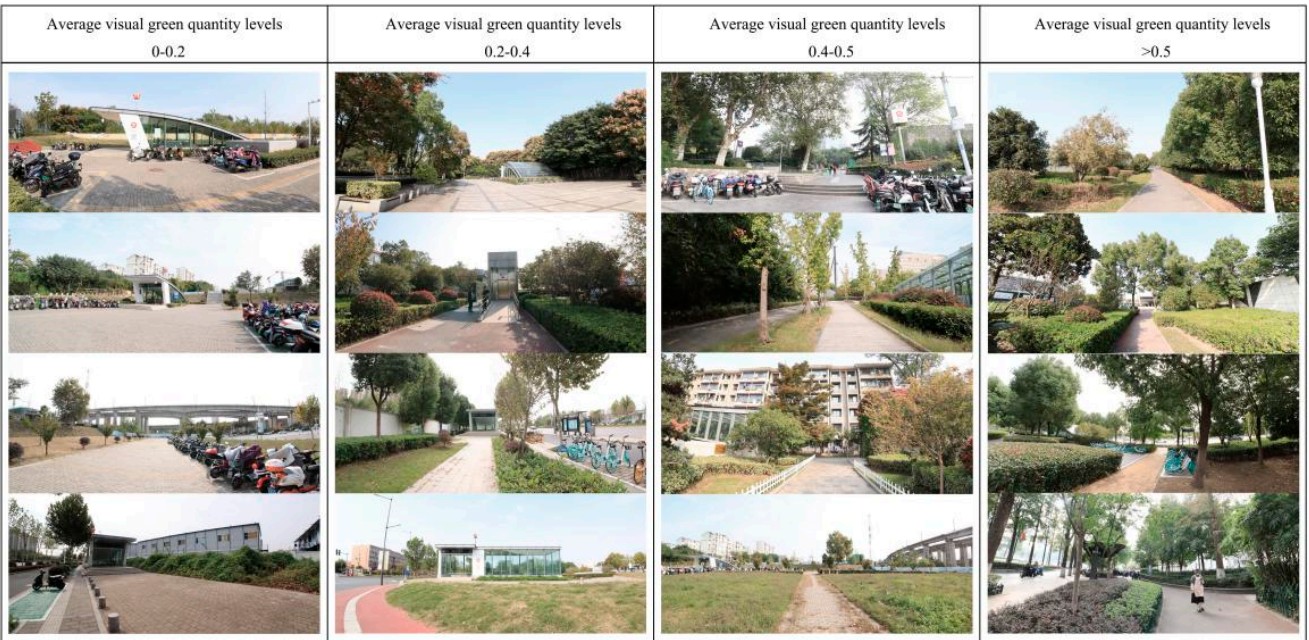

**Figure 7.** Examples of acquisition results of different average visual green quantity levels.

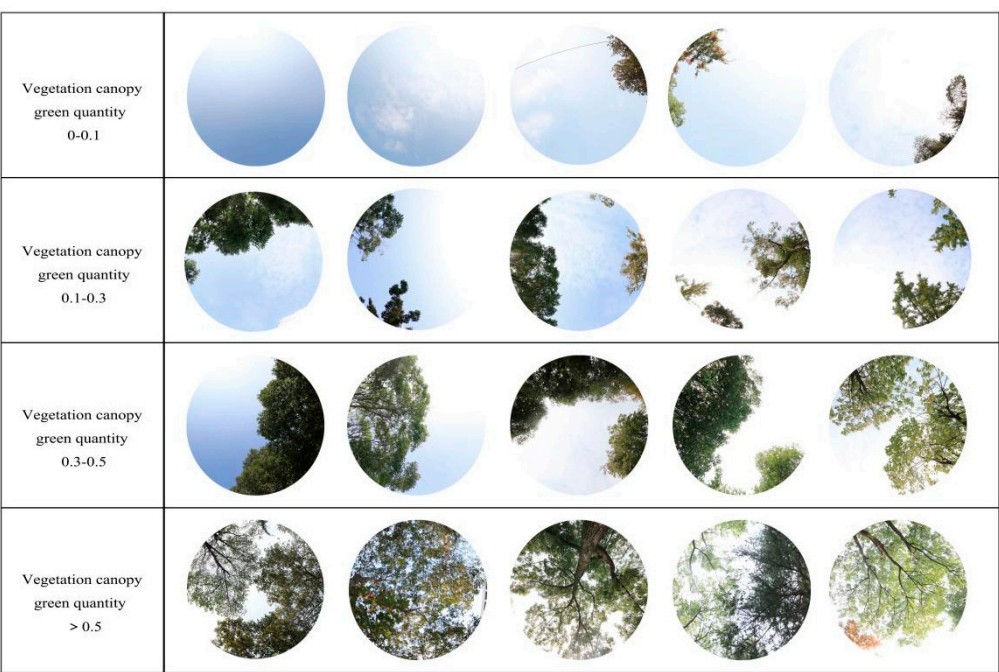

**Figure 8.** Sample of green quantity collected at different vegetation canopy levels.

## 5. Efficiency Structural Equation Model

### 5.1. Factor Analysis

Before importing data into the theoretical model, it is important to test the validity and rationality of the selected variables. The normalized data were imported into spss24.0 software for the exploratory factor analysis. The results of the Kaiser–Meyer–Olkin test and the significance test (<0.05) show that the observed variables are correlated. To avoid the meaningless explication of factors, component fit testing was conducted with the principal component analysis method. As shown in Table 3, the component extraction results of the latent variables "landscape facility service ability" and "green space landscape" are completely consistent with the pre-set theoretical model. Since the coefficients are higher than 0.9 and the construct reliability result is also ideal, this paper concluded that the latent variables are interrelated with the corresponding observation variable group [45].

**Table 3.** Validity test of evaluation factors.

| Latent Variable | Observation Variable | Factor Loading | Cronbach A Coefficient | Average Variance Extracted | Construct Reliability |
|---|---|---|---|---|---|
| Service capacity of landscape facilities | Density of recreational facilities<br>Accessibility coverage<br>Pavement integrity of pedestrian space<br>Bearing capacity of parking facilities | 0.654<br>0.933<br>0.940<br>0.944 | 0.942 | 0.7682 | 0.9285 |
| Green space landscape | Plant community species density<br>Shading rate of walking space<br>Average visual green quantity<br>Vegetation canopy green quantity | 0.754<br>0.844<br>0.785<br>0.864 | 0.918 | 0.6609 | 0.886 |

### 5.2. Parameter Estimation

In order to verify whether the measurement relationship between variables is established, parameter fitting analysis is required. If the coefficient results are not ideal, researchers can define whether to add or delete redundant parameter paths by the observation of fitting indicators. In the assessment of a model, some studies recommend applying the maximum likelihood method when the data distribution meets the principle of normality, and the number of samples exceed 100. After the normality test, the results show that the skewness coefficient and kurtosis coefficient of all observation variables are

less than 1.5, which conforms to the law of normal distribution. Since the sample size of this study is 131, it is suitable to use the maximum likelihood method for parameter estimation.

The normalized data were imported into the proposed theoretical model by Amos software. The model fitting results show that the latent variables (landscape facility service capacity and green space landscape) directly affect the landscape efficiency of subway station entrances with high factor loading (Figure 9). Similarly, the observed variables (connectivity and spatial integration) correlate with landscape efficiency to different degrees. In other words, previous theoretical assumptions have achieved the parameter estimation, while the validity of the model structure still needs confirmatory analysis.

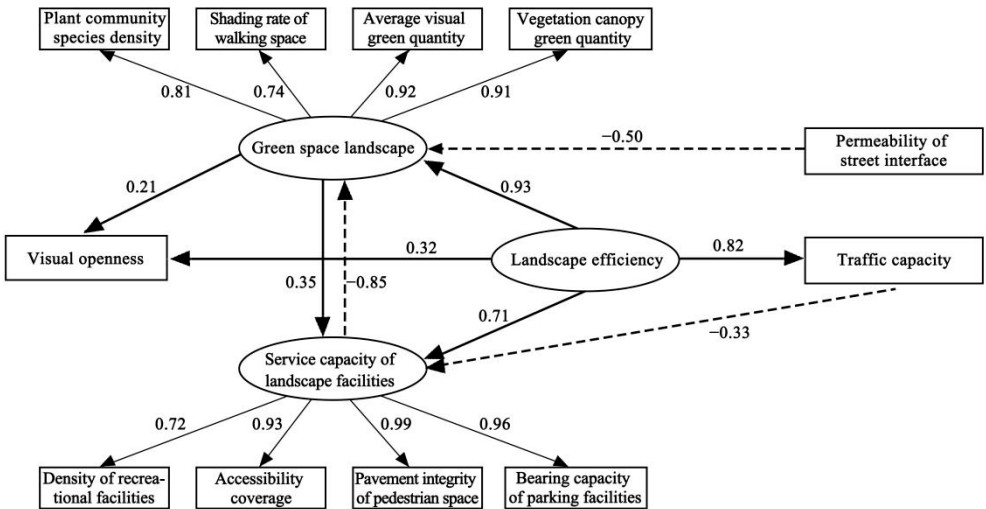

**Figure 9.** Fitting results of subway station entrance landscape efficiency Model (obtained by AMOS).

### 5.3. Model Fitting Test

The purpose of fitness evaluation for the subway station entrance landscape efficiency model is to develop a latent variable model with good fitness in theory and statistics. The fitness test of model includes three types of evaluation indices [46]: absolute fit indices, relative fit indices and parsimonious fit indices. In testing, it is necessary to comprehensively judge the fitting status of the model regarding the evaluation results of different indexes.

In the evaluation of absolute fit indices, the comparative fit index (CFI) is 0.927, which meets the standard requirements that should be greater than 0.9. The root mean square error of approximation (RMSEA) is 0.075, which is slightly less than the critical value of 0.08 [47]. Among relative fit indices, both incremental fit index (IFI) and rule adaptation index (PNFI) meet the adaptation standard, and only the standard fit index (NFI) is slightly lower than the standard of 0.9 [48]. The evaluation results of the parsimonious fit indices are also within the critical value. In conclusion, the model passes the confirmatory analysis since it basically meets the test criteria (Table 4).

**Table 4.** Suitability evaluation of subway station entrance landscape efficiency model (obtained by AMOS software analysis).

| | Absolute Fit Index | | Value Added Adaptation Index | | | Simple Adaptation Index | |
|---|---|---|---|---|---|---|---|
| Adaptation parameter | CFI | RMSEA | NFI | IFI | PNFI | NC value | PCFI |
| Adaptation threshold | >0.9 | <0.08 | >0.9 | >0.9 | >0.5 | 1 < NC < 3 | >0.5 |
| Research results | 0.927 | 0.075 | 0.851 | 0.929 | 0.674 | 1.770 | 0.674 |

### 6. Analysis and Discussion

According to the path diagram (Figure 9), the latent variable "landscape efficiency" is positively correlated with the service capacity of landscape facilities, green space landscape,

spatial integration and connectivity to varying degree. Besides, there are complex intrinsic relationship among these impact factors which might change the design strategies in the specific subway station entrance. Therefore, the estimation results have to be discussed in four aspects.

(1) Initially, the results imply that the factors with significant benefits for landscape efficiency are the service capacity of landscape facilities and the landscape of green space. This finding verifies that the traffic demand is of the priority in landscape design of subway station entrances. Similarly, green space also has an overwhelming influence on landscape efficiency, which might stem from the ornamental and ecological benefits of plants. We believed that rich plant landscape construction and appropriate traffic layout can significantly improve the landscape efficiency of subway station entrances. It can help subway station entrances to enhance their spatial guidance capacity, improve tourist satisfaction, and act as a landscape hub. In contrast, the intrinsic relationship between the service capacity of landscape facilities and the landscape efficiency of subway station entrances is slightly weaker, which may be caused by the excessively simple content of landscape facilities. Although visual openness has relatively weak effects in landscape efficiency, it cannot be ignored in landscape design. The change in any latent variable may cause the related latent variable to decay or enhance, and then have a dynamic, nonlinear and complex impact on the overall efficiency of the landscape space. In conclusion, micro-environmental factors play a more significant role in landscape efficiency, because they can affect landscape space more directly.

(2) There is a complex correlation mechanism among these variables, presenting different tendencies such as restriction or promotion. Any latent variable does not exist independently in the subway station entrance landscape system. There are interactions between these latent variables and one of them is weakened; other landscape elements will change simultaneously. Estimation results infer that the factor "traffic capacity" determines the landscape efficiency of the subway station entrance without the influence of space scale to some extent. Additionally, it has a certain limiting effect on the latent variable "landscape facility service capacity". Landscape facility service capacity and green space landscape have a mutual relationship. The landscape facility service capacity has a strong negative impact on the green space landscape, while the latter can affect the former positively. In landscape design, it is necessary to comprehensively consider the specific situation and determine whether to sacrifice the service capabilities of landscape facilities appropriately. For the subway station entrances that are mainly traffic-oriented or have weak commercial influences, such as the entrances of Fuqiao Station, the focus of improving the aesthetic value and comfort of green landscapes is acceptable. Moreover, there is a weak correlation between green space landscape and visual openness. This is probably a consequence that the plants are the main three-dimensional obstacles to restrict the visual openness (namely, the visual level).

(3) For the latent variable "landscape of green space", it is noteworthy that the interface permeability has an inhibitory effect on the latent variable "green space landscape". Therefore, it is essential to avoid the excessive complexity of the green space landscape at the subway station entrance. Meanwhile, green space design should consider the connection with the surrounding urban green space. The path analysis of latent variables further shows that the landscape of green space is a comprehensive description of observed variables (shading rate of pedestrian space, species density of plant community, green quantity of vegetation canopy and average visual green quantity). The four observed variables have a positive impact on the landscape of green space. Consequently, designers should establish a cognition that the green space is an integrity. It is difficult to optimize a specific element without seizing the coupling relationship between greening elements. For example, to improve the visual perception of plants, it is necessary to increase greening layers and enrich plant

varieties. It is recommended to plant large trees in the inner activity space, then adjust the spatial allocation of plant trees and shrubs near pavements.

(4) The service capacity of landscape facilities is the superposition effect of the various landscape facilities at the subway station entrance. Comparing path loading, it can be found that the service efficiency of landscape facilities has strong correlations with the pavement integrity and bearing capacity of parking facilities. Since users visit the station mainly on foot or by bike, providing excellent pavement quality and reasonably arranged bicycle parking space can ensure efficient traffic process and promote user satisfaction. However, the density of recreational facilities and accessibility coverage have weak effects on landscape facility service capacity. The field survey results also show that visitors need fewer recreation facilities. In subway station entrances with large traffic volume (such as Xinjiekou Station, Fuzimiao Station and Gulou Station); researchers suggest reducing seats to restrict the duration of resident activities and enhance the efficiency of pedestrian flow. However, for subway station entrances with less foot traffic, which are an important activity space for residents, it is inadvisable to reject the implantation of recreational functions.

## 7. Conclusions and Prospects

This study analyses and evaluates the correlation between the landscape efficiency of subway station entrances and four latent variables: landscape facility service capacity, green space landscape, spatial integration and connectivity by the structural equation model method. Based on the parameter estimation, we complete a confirmatory analysis of the evaluation system and discuss the design strategies of subway station entrances according to the path diagram. With the support of sample data, this paper summarises the typical characteristics of subway station entrances in the downtown area of Nanjing and forms an intuitive description of landscape efficiency.

Compared with previous research, this paper presents improvement in three aspects: research object, technical application and theoretical verification. In terms of research type, it is a breakthrough in incorporating the subway station entrance into the research content of landscape efficiency. Researchers concentrate on the analysis of physical environment elements, which also concerns the systematic organization form of landscape space. In the observed data collection, the spatial identification methods in relevant research fields such as statistics, architecture and urban planning are integrated to form a multivariate data collection method for miniature public space. Furthermore, this paper attaches great importance to the derivation of theoretical models and establishes a compound path system instead of a single path. A systematic latent variable analysis technology is formed in the construction of a theoretical model, screening factors, optimization path, fitness verification and even a measurement method. Regarding practical application, the results can be used to carry out retrospective analysis on landscape design and help designers identify design defects. However, it should be noted that there are some limitations to this study. Since the research scope is the main urban area of Nanjing, it cannot be ensured that the conclusion is universal to other cities. The factor sets of landscape efficiency need to be expanded, adding some indicators related to the behaviour patterns of pedestrians. Notwithstanding its limitations, this study does provide a scientific basis for proposing design strategies of subway station entrances. In future study, it is suggested to apply the results to more projects and increase the sample library in other cities.

It is one of the important symbols of the contemporary development of landscape architecture to apply digital technology in landscape efficiency research. To improve the landscape environment, researchers should adopt mathematical models with a parameter system, strengthen logical analysis and improve estimation methods. This paper applies structural equation model technology to establish a universal research framework of landscape efficiency and explore the ideal landscape construction forms. The results can be applied to the research of other micro landscape spaces to improve the scientific

cognition of landscape efficiency. Additionally, it can promote construction effects matching design expectation and help to build a high-quality landscape environment.

**Author Contributions:** Conceptualization, Z.L. and X.L. (Xiaoshan Lin); Data curation, H.Z.; Funding acquisition, Z.L.; Investigation, X.L. (Xiaoshan Lin) and X.L. (Xinyi Lu); Methodology, Z.L.; Resources, X.H.; Software, X.L. (Xiaoshan Lin); Supervision, X.L. (Xinyi Lu); Validation, X.H., X.L. (Xinyi Lu) and H.Z.; Visualization, X.L. (Xiaoshan Lin) and X.L. (Xinyi Lu); Writing—original draft, Z.L.; Writing—review and editing, X.L. (Xiaoshan Lin). All authors have read and agreed to the published version of the manuscript.

**Funding:** This research was funded by the National Key R&D Program of China, grant number [2019YFD1100405] and its branch project, grant number [2019YFD11004055].

**Data Availability Statement:** The data presented in this study are available on request from the corresponding author. The data are proprietary or confidential in nature and may only be provided with restrictions.

**Conflicts of Interest:** The authors declare no conflict of interest.

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
