# Peer review of "Landscape Efficiency Assessment of Urban Subway Station Entrance Based on Structural Equation Model: Case Study of Main Urban Area of Nanjing"

_buildings, doi:10.3390/buildings12030294_

Round 1

Reviewer 1 Report

Please, 

see the attached file.

Kind Regards

Author Response

Dear Reviewer:

Thank you for your comments and advice for this article. We have made structural adjustments to the overall paper, revised the confusing content in the discussion, corrected some spelling and grammatical errors, and optimized the diagrams.

Please download the attachment and revised manuscript.

Best regards.

Reviewer 2 Report

The number of stations and the evaluated factors are representative for the approached subject. The paper supports in an appropriate graphic manner the research methods and their results. The discussions analyze the results extensively and are correlated with the conclusions. The combination of various methods of analysis: visual analysis, digital technology, statistical analysis has led to scientifically validated results related to landscape efficiency assessment.

Author Response

(The authors gave the same response as above.)

Reviewer 3 Report

This paper is fascinating, the research would show the deep theory of urban landscape, yet some notes should be considered. 
The Abstract's result does not show the aims of the research in detail also the methodology of the research does not show clearly. 
The authors should show the methodology, how did they obtain the data, for instance, how did they generate the visual data of the vegetation canopy (vegetation Canopy) (is it like the Sky View factor/SVF?), where did they put the camera to obtain the value of VC. It has many possibilities of VC value depending on where you put the camera. 
Table 1 should add one column on how the data were obtained, what software was used, how to analyze it. 
Figure 5, how to calculate this value. 
The result of this paper still has confusing sentences. 

Author Response

(The authors gave the same response as above.)

Round 2

Reviewer 3 Report

Ok, I have read, everything is going to be fine.